# Immunotoxin Screening System: A Rapid and Direct Approach to Obtain Functional Antibodies with Internalization Capacities

**DOI:** 10.3390/toxins12100658

**Published:** 2020-10-15

**Authors:** Shusei Hamamichi, Takeshi Fukuhara, Nobutaka Hattori

**Affiliations:** 1Research Institute for Diseases of Old Age, Juntendo University School of Medicine, Tokyo 113-8421, Japan; s.hamamichi.mo@juntendo.ac.jp; 2Department of Neurology, Juntendo University School of Medicine, Tokyo 113-8421, Japan; nhattori@juntendo.ac.jp; 3Department of Research for Parkinson’s Disease, Juntendo University Graduate School of Medicine, Tokyo 113-8421, Japan; 4Neurodegenerative Disorders Collaborative Laboratory, RIKEN Center for Brain Science, Saitama 351-0198, Japan

**Keywords:** monoclonal antibody, immunotoxin, antibody drug conjugate, immunoliposome, drug delivery, diphtheria toxin, DT3C

## Abstract

Toxins, while harmful and potentially lethal, have been engineered to develop potent therapeutics including cytotoxins and immunotoxins (ITs), which are modalities with highly selective targeting capabilities. Currently, three cytotoxins and IT are FDA-approved for treatment of multiple forms of hematological cancer, and additional ITs are tested in the clinical trials or at the preclinical level. For next generation of ITs, as well as antibody-mediated drug delivery systems, specific targeting by monoclonal antibodies is critical to enhance efficacies and reduce side effects, and this methodological field remains open to discover potent therapeutic monoclonal antibodies. Here, we describe our application of engineered toxin termed a cell-based IT screening system. This unique screening strategy offers the following advantages: (1) identification of monoclonal antibodies that recognize cell-surface molecules, (2) selection of the antibodies that are internalized into the cells, (3) selection of the antibodies that induce cytotoxicity since they are linked with toxins, and (4) determination of state-specific activities of the antibodies by differential screening under multiple experimental conditions. Since the functional monoclonal antibodies with internalization capacities have been identified successfully, we have pursued their subsequent modifications beyond antibody drug conjugates, resulting in development of immunoliposomes. Collectively, this screening system by using engineered toxin is a versatile platform, which enables straight-forward and rapid selection for discovery of novel functional antibodies.

## 1. Introduction

Immunotoxin (IT), a subgroup of immunoconjugates, consists of a target recognition moiety that is linked to bacterial or plant proteineous toxins [1,2]. As an IT, the target recognition moiety is a full-length monoclonal antibody or antibody fragment that specifically binds to an antigen expressed on the surface of target cell, and as a cytotoxin, the component includes a receptor-specific ligand, such as cytokine, chemokine receptor ligand, and growth factor [3,4]. The cytotoxic protein is composed of a toxin derived from bacteria, such as *Pseudomonas aeruginosa* exotoxin A (PE) or diphtheria toxin (DT), as well as from plants including ricin, saporin, gelonin, and bouganin [5,6,7,8,9,10]. While simple in conceptual design, consisting of two major components, multiple combinations of these two parts allow unlimited prospects to generate potential therapeutic agents with target selectivity. As conceived by Paul Ehrlich with his “magic bullet” concept [11], various types of ITs epitomize potential therapeutic agents with capacities to target disease-relevant antigens.

Current challenges for development of IT as a therapeutic agent include immunogenicity and stability of the fusion protein as well as binding affinity of the target recognition moiety [12]. Here, we overview current toxin-mediated therapeutics, and focus on the target recognition moiety; i.e., monoclonal antibody. Additionally, there are a growing number of highly effective antibody-mediated therapeutics, such as antibody drug conjugates (ADCs). Therefore, we revisit antibody generation technology, beginning from the monumental work on development of the hybridoma technology reported by Köhler and Milstein in 1975 [13], for which they were awarded the Nobel Prize in Physiology and Medicine in 1984. Since then, various advancements for high throughput production of these antibodies have been proposed and reported [14,15]. Here, we compare multiple screening systems to obtain monoclonal antibodies, and describe our unique strategy termed a cell-based IT screening system. The IT screening system, which utilizes distinct features of antibody and engineered toxin, is a rapid, and perhaps more importantly, direct method to identify antibodies that recognize cell surface molecules and are internalized into the cells to induce cytotoxicity. In principle, the selected antibodies through this screening system are suitable for ADCs, immunoliposomes (ILPs) or other drug delivery systems.

## 2. Current FDA-Approved Toxin-Mediated Therapeutics

Presently, three toxin-mediated therapeutics, such as cytotoxins and IT have been approved by the U.S. Food and Drug Administration (FDA) (Table 1). Denileukin diftitox (Ontak^®^), administered as an antineoplastic agent for treatment of persistent or recurrent cutaneous T-cell lymphoma, is comprised of a full-length sequence of IL2 protein that is fused to truncated DT (DAB389) [16]. This fusion protein is targeted to the cells expressing interleukin-2 receptor (IL2R), and upon binding, denileukin diftitox is internalized by receptor-mediated endocytosis and proteolytically cleaved to generate a fragment of DT that inhibits protein synthesis by ADP-ribosylation of elongation factor (EF)-2 and induces cytotoxicity [17]. Tagraxofusp (Elzonris^®^), used for treatment of blastic plasmacytoid dendritic cell neoplasms, is composed of a human IL3 protein and truncated DT [18]. Moxetumomab pasudotox (Lumoxiti^®^), approved for treatment of relapsed or refractory hairy cell leukemia, consists of a binding fragment (Fv) of anti-cluster of differentiation-22 (CD22) antibody (RFB4) and a 38 kDa portion of PE termed PE38 [19]. Currently, over 20 IT therapeutics are being tested in the clinical trials. As elegantly reviewed by Kim et al. [20], common themes among the FDA-approved toxin-mediated therapeutics include the target recognition moiety that specifically targets hematological cancer cells, and the truncated bacterial toxins that allow reduced levels of immunogenicity and non-specific binding.

In addition to the current FDA-approved therapeutics, multiple toxin-mediated modalities targeting solid tumors are presently in the clinical trials. These modalities include cintredekin besudotox (IL13-PE38QQR) for glioblastoma [28], oportuzumab monatox (VB4-845) for urothelial carcinoma [29], naptumomab estafenatox for renal cell carcinoma [30], and LMB-100 for advanced pancreatic adenocarcinoma [31]. At the preclinical level, ITs have been developed and modified to target activated macrophages [32], murine noradrenergic neurons in the locus ceruleus [33], and human immunodeficiency virus (HIV)-infected cells [34]. While most previous and present ITs and cytotoxins have been designed to target cancer, as long as target selectivity and cytotoxicity are desired, these modalities can conceptually be applied for treatment of various diseases including neuronal diseases.

For future generation of ITs with enhanced efficacies and reduced adverse events, improved target recognition and reduced immunogenicity are key factors. The latter topic, which focuses on immunogenicities associated with antibodies, and especially bacterial and plant toxins are reviewed elsewhere [35,36,37] and beyond the scope of this review article. Improvement on target recognition is dependent upon specificity and affinity of monoclonal antibody. Furthermore, given the necessity to deliver the toxin into the cell, internalization of the antibody is pivotal. Regardless of the major advancements, such as humanized ITs [38,39] or bispecific ITs targeting CD19 and CD22 [40,41], quality of the antibody unequivocally remains crucial. Therefore, we must carefully employ an antibody screening strategy that maximizes the attainment of the elite antibody with sufficient internalization capacity suitable for subsequent modification as an IT, ADC or ILP to deliver the payload like toxin. Simultaneously, it is also important to identify disease-specific antigens, especially epitopes.

## 3. Mode-of-Actions of Therapeutic Antibodies

It is important to emphasize that not all antibody-based therapeutics require internalization capacities since therapeutic efficiencies are dependent on functionalities of the antibodies [42,43]. Monoclonal antibodies, such as rituximab (Rituxan^®^; a chimeric anti-CD20 monoclonal antibody) and trastuzumab (Herceptin^®^; a humanized anti-HER2 monoclonal antibody) induce cytotoxicities of target cells via antibody dependent cellular cytotoxicity (ADCC) [44,45,46]. Through conjunction with radionuclides, such as ^111^In or ^90^Y, ibritumomab tiuxetan (Zevalin^®^; a mouse anti-CD20 monoclonal antibody) is a radioimmunoconjugate with both diagnostic and therapeutic (i.e., theranostic) capacities [47,48]. Bevacizumab [Avastin^®^; a humanized anti-vascular endothelial growth factor-A (VEGF-A) monoclonal antibody] neutralizes a ligand, which consequently reduces microvascular growth [49,50]. Nivolumab [Opdivo^®^; a human anti-programmed cell death protein 1 (PD-1) monoclonal antibody] and pembrolizumab (Keytruda^®^; a humanized anti-PD-1 monoclonal antibody) block PD-1 expressed on the lymphocytes, thus allowing the immune cells to attack cancer cells through modulation of the immune system [51,52]. Currently in Phase 3 clinical trial, photoimmunotherapy is another promising approach wherein a monoclonal antibody is conjugated with IRdye700DX, and localized exposure to near-infrared (NIR) light activates the switch that results in rapid and selective death of targeted cancer cells [53,54]. These are a few among many therapeutic monoclonal antibodies that are not necessary to be internalized, but since our focus includes delivery of the toxin into the cells, internalization capacity cannot be ignored.

## 4. Internalization Assays to Obtain Monoclonal Antibodies with Internalization Capacities

In case of hybridoma technology, production of monoclonal antibody generally starts with immunization of animals with antigens, followed by isolation of splenocytes. Splenocytes and myeloma cells are fused together to generate hybridomas, and the culture supernatants of these hybridomas need to be screened to identify the candidate antibodies. Subsequent process with limiting dilution enables to establish a hybridoma clone that produces monoclonal antibody. In general, antibody recognizes a structure of corresponding region that consists of approximately 15 amino acids, suggesting that the selection of antibody is affected by the screening methods due to the structural state of target molecule. In other words, the selected antibodies are primarily suitable for the screening methods used. Common assays include immunocytochemistry with fixed cells, enzyme-linked immunosorbent assay (ELISA) with recombinant proteins, and immunoblotting with denatured proteins, but these methods are not always guaranteed to select the potent therapeutic antibodies that recognize the in vivo states of antigens. While these techniques offer notable advantages for selecting antibodies with different characteristics, these procedures are not suitable to determine internalization properties of the antibodies, which are indispensable for most types of drug delivery systems. Therefore, it should be reminded that candidate antibodies have to be screened with consideration for the in vivo structural states of antigenic molecules expressed on the target cells.

As for previous efforts to determine internalization of monoclonal antibody, reported methods are categorized as either direct internalization or indirect internalization assays. Direct internalization assay generally utilizes a purified primary antibody labeled with radioisotope [55] or fluorescence probe [56,57]. These methods possess two obstacles including a necessity to purify primary antibody (hence, supernatants directly obtained from the hybridoma library cannot be used), and a facility that allows the use of radioisotopes or fluorescence scanner. The former issue can be solved by performing indirect internalization assay wherein a purified secondary antibody is labeled, but the latter depends on the facility infrastructure. Additionally, while cellular internalization may sufficiently be measured, subsequent cytotoxicity is not analyzed; hence, internalization and cytotoxicity of IT is more accurate evaluation to assess the potential of the antibody for drug delivery. Additionally, it is feasible to conduct flow cytometry to identify receptor internalization by analyzing the cells with or without PMA stimulation; however, this approach is not suitable for high-throughput screening [58].

Internalization and cytotoxic properties of ITs have been reported through direct labeling with radioisotope [59] or fluorescent dye with functional group [60]. These techniques require purified ITs; therefore, they are inappropriate as a screening procedure to distinguish the hybridomas that secrete desired antibodies from those that do not produce them. More favorable approach is indirect IT assay wherein a toxin-labeled secondary antibody is utilized to identify a monoclonal antibody of interest [61,62]. While successful, since this procedure relies on the toxin-labeled secondary antibody, structural composition is not identical to the final form of IT or ADC where monoclonal (primary) antibody is directly linked to toxins or drugs. Taken together, these issues highlight a major demand to establish an IT screening system with more predictive analysis of the monoclonal antibody.

## 5. Cell-Based Immunotoxin Screening System

To this end, we previously reported a cell-based IT screening system to facilitate the identification and isolation of monoclonal antibodies with internalization properties [63]. This rapid and direct screening strategy offers the following advantages: (1) identification of monoclonal antibodies that recognize cell-surface molecules, (2) selection of the antibodies that are internalized into the cells, (3) selection of the antibodies that induce cytotoxicity since they are linked with toxins, and (4) determination of state-specific activities of the antibodies by differential screening under multiple experimental conditions. Unlike the procedures described above, our cell-based IT screening system does not require radioisotopes nor fluorescent dyes, and it does not utilize toxin-labeled secondary antibodies. Collectively, our approach provides a platform for direct discovery of potent IT-, ADC- or ILP-compatible antibodies in vitro.

Prior to the IT screening, our approach undergoes three major steps to generate a hybridoma library: (1) immunization, (2) cell-fusion of splenocytes and myeloma cells, and (3) library construction of hybridomas through hypoxanthine-aminopterin-thymidine (HAT) selection. Generation of the hybridoma library is advantageous since hybridoma technology allows continuous growth of hybridomas and production of a large quantity of purified antibodies by conventional method [64]. Additionally, the hybridoma library can be frozen and stored for later studies. To conduct the IT screening, the supernatants from the hybridoma library were pre-incubated with engineered toxin DT3C to form ITs (Figure 1a). DT3C is a recombinant protein that consists of DT without the receptor-binding domain but containing the fragment crystallizable (Fc)-binding domain of *Streptococcus* protein G (3C) [65]. Because of this remarkable feature, DT3C specifically binds to an antibody with affinity. As summarized in Figure 1b, if the IT recognizes a surface molecule expressed on the cell, then the IT is internalized wherein DT3C is cleaved by the cytosolic furin protease, and catalytic domain of DT3C is released into the cytoplasm. The released catalytic domain leads to ADP-ribosylation of EF-2, followed by inhibition of the protein translation machinery and ultimately cytotoxicity.

Utilizing the unique principle of our IT assay system, we performed primary screening to identify antibody-secreting hybridomas that were capable of inducing DT3C-dependent cytotoxicity (Figure 1c). Induction of cytotoxicity was assessed by WST-1 assay to quantitatively evaluate cell viability. We established 90G4 clone that produced functional monoclonal antibody with capacity for DT3C-dependent cytotoxicity [63]. Subsequently, 90G4, a rat anti-mouse CD321/F11 receptor antibody was further characterized by flow cytometry, immunocytochemistry, immunoprecipitation, immunoblotting, and mass spectrometry, demonstrating differential expression patterns of CD321 under normoxic vs. hypoxic conditions. This study revealed new roles of endothelial CD321 that was internalized upon the hypoxic signal. While we utilized the hybridoma technology to generate monoclonal antibodies, it is noteworthy that, in principle, our cell-based IT screening system is applicable to the screening of the antibodies obtained from antibody phage display [66,67,68] and single B cell antibody technologies [69,70].

## 6. Application of Functional Monoclonal Antibodies as Antibody Drug Conjugates

ADCs have gained significant attention as highly potent therapeutic agents because of their pharmacological characteristics including target specificity, target-binding affinity, good retention, and low immunogenicity that altogether contribute to targeted drug delivery and decreased side effects [71,72,73,74]. Accordingly, FDA-approved ADCs include gemtuzumab ozogamicin (Mylotarg^®^; a humanized anti-CD33 monoclonal antibody conjugated to ozogamicin) for CD33-positive acute myeloid leukemia [21,75]; brentuximab vedotin [Acetris^®^; a chimeric anti-CD30 monoclonal antibody conjugated to monomethyl auristatin E (MMAE)] for relapsed Hodgkin lymphoma, systemic anaplastic large cell lymphoma, and other CD30-expressing peripheral T-cell lymphomas [22,23]; trastuzumab emtansine (Kadcyla^®^; a humanized anti-HER2 monoclonal antibody conjugated to DM1) for HER2-positive breast cancer [24,25]; and inotuzumab ozogamicin (Besponsa^®^; a humanized anti-CD22 monoclonal antibody conjugated to ozogamicin) for acute lymphoblastic leukemia [26] (Table 1). Recently, four additional ADCs have been FDA-approved; polatuzumab vedotin (Polivy^TM^; a humanized anti-CD79B monoclonal antibody conjugated to MMAE) for diffuse large B cell lymphoma, enfortumab vedotin (Padcev^TM^; a human anti-nectin-4 monoclonal antibody conjugated to MMAE) for urothelial cancer, trastuzumab deruxtecan (Enhertu^®^; a humanized anti-HER2 monoclonal antibody conjugated to deruxtecan) for unresectable or metastatic HER2-positive breast cancer, and sacituzumab govitecan (Trodelvy^TM^; a humanized anti-Trop-2 monoclonal antibody conjugated to SN-38) for metastatic triple-negative breast cancer [27].

Typically, ADC consists of a monoclonal antibody that is linked to cytotoxic payloads by non-cleavable or cleavable linker [71,72,73,74]. Similar to IT, quality of the monoclonal antibody is crucial because therapeutic property of ADC is partially related to the characteristics of its antigen. Specifically, not only are differential surface expression levels of antigens between the target and non-target cells essential, the antigens are preferred to possess internalization properties because it will facilitate the ADC to be transported into the cells and enhance its efficacy. The notion was also illustrated for generation of ADC by using a human single chain variable fragment (scFv)-Fc antibody against CD239/basal cell adhesion molecule for treatment of breast cancer [76]. The scFV-Fc antibody termed C7-Fc, originally identified from screening the scFv phage libraries [77], was bound to DT3C to characterized its internalization property, and selectively target and kill SKBR3 breast cancer cells. This work also provides evidence supporting the application of our cell-based IT screening system to characterize the candidates identified from the phage display libraries.

From the IT screening method, a mouse anti-human CD71/transferrin receptor antibody termed 6E1 antibody was identified and isolated [65]. To assess its cytotoxic potential, purified 6E1 antibody was pretreated with DT3C and administered to A172, SH-SY5Y, and H4 cells. As an IT, 6E1:DT3C demonstrated strong cytotoxic activities wherein logEC50s (ng/mL) were 4.59 (A172), 2.27 (SH-SY5Y), and 6.87 (H4) (Figure 2). To generate its ADC form, antibody binding peptide termed Z33 was elegantly conjugated with anti-cancer agent plinabulin, and Z33-conjugated plinabulin was then used to non-covalently bind to 6E1 antibody [78]. As expected, this ADC demonstrated enhanced cytotoxicity against CD71-positive melanoma A375 cells. In addition to the 6E1 antibody, mouse anti-human Mucin 13 (MUC13) antibody termed TCC56 was identified from the IT screening method and shown to induce cell death in TCC-PAN2 cells expressing MUC13 [79].

## 7. Challenges for Next Generation of Antibody Drug Conjugates

Optimization of monoclonal antibody, linker, and cytotoxic payload, as well as increased retention and enhanced penetration to the target cells are key factors associated with improved next generation of ADCs that will enhance efficacies and reduce side effects [71,72,73,74]. Moreover, one of the critical features to consider for development of ADCs as drug delivery modalities is drug-to-antibody ratio (DAR). For instance, high DAR affects antibody structure and stability, and low DAR can decrease efficacy; therefore, in most cases, their DAR values are restricted to the manageable ranges [80,81]. Recently, Ogitani et al. reported that a HER2-targeting ADC termed DS-8201a was successfully conjugated with 8 molecules of novel topoisomerase I inhibitors per antibody, and that this ADC exhibited potent anti-tumor activities in a wide range of HER2-positive animal models with favorable pharmacokinetics and safety profiles [82]. To maximize drug load and minimize structural changes and instability of the monoclonal antibody, one avenue to explore is generation of ILP.

## 8. Application of Functional Monoclonal Antibodies as Immunoliposomes

There are numerous advantages to liposomes, such as biocompatibility, biodegradability, low toxicity, and their capacities to encapsulate both hydrophilic and hydrophobic drugs [83,84]. Conventional liposomes are phospholipid bilayers that are composed of certain molar ratios of phospholipids and cholesterols with drugs entrapped inside. To prolong blood circulation time and reduce uptake by the cells of the reticuloendothelial system, polyethylene glycol (PEG) was added to the surface of the liposomes termed “stealth” liposomes. Consequently, multiple formulations of conventional and PEGylated liposomes have been FDA-approved including Doxil^®^, DaunoXome^®^, Depocyt^®^, Marqibo^®^, Onivyde^®^ and Vyxeos^®^ for various forms of cancer, as well as Amphotec^®^ and Ambisome^®^ for fungal infections [85,86,87,88]. Potential candidates for liposomal encapsulation include clinically approved or presently developed therapeutic agents for treatment of cancer, neurodegenerative diseases, cardiovascular diseases, inflammation, and infections [89,90]. Moreover, combined with imaging modalities, such as magnetic resonance imaging (MRI), ultrasound, single-photon emission computed tomography (SPECT), and positron emission tomography (PET), MRI- or ultrasound-contrast agents, as well as radionuclides can be encapsulated into the liposomes for diagnostic, therapeutic, and potentially theranostic application [91,92,93,94].

To introduce novel functionalities to the liposomes, monoclonal antibodies or their fragments, such as fragment antigen-binding (Fab) or scFv, are conjugated to the liposomal surface to generate ILPs [95,96]. Given the clinical success of Doxil^®^ (PEGylated liposome encapsulating anti-cancer agent doxorubicin) [97], next endeavor was active targeting to attain improved drug delivery and therapeutic efficacy. Park et al. generated anti-HER2 ILP that encapsulated doxorubicin, and reported identical prolonged blood circulation compared with control PEGylated liposome without antibody conjugation, as well as enhanced therapeutic outcomes in four different HER2-overexpressing tumor xenograft models [98]. Furthermore, anti-HER2 ILP that encapsulated paclitaxel and rapamycin also demonstrated controlled tumor growth in a mouse orthotopic HER2-positive SKBR3 xenograft model [99]. Conversely, a lack of increased anti-tumor activity after ILP administration has also been reported [100]. Considering the heterogenous and complex nature of cancer, drug delivery, tumor accumulation of the ILPs, internalization of the encapsulated drugs, and multiple additional factors combined contribute to therapeutic efficiencies [101,102,103,104]. Collectively, ILP is a bifunctional modality that possesses the qualities of antibody and liposome with unequivocal potential for targeting specificity and delivery of immensely encapsulated drugs.

In addition to development of cancer therapeutics, at the preclinical level, another intriguing concept is delivery of drugs across the blood brain barrier (BBB) into the brain by using ILPs. The BBB, where cell membrane proteins involved in receptor-mediated transcytosis, such as transferrin receptor and insulin receptor are expressed, is a selective barrier between systemic blood circulation and brain parenchyma [105,106,107,108]. Recently, Johnsen et al. reported that intravenous injection of OX26 (a mouse anti-rat CD71/transferrin receptor monoclonal antibody)-conjugated and oxaliplatin-encapsulated liposome resulted in higher concentration of platinum in the rat brain parenchyma compared to the control ILP [109]. We have also tested the application of functional monoclonal antibody as an ILP whereby purified 6E1 antibody was conjugated with DiOC_18_(3)-encapsulated liposome, and the resulting ILP was applied to A172 cells. We observed 5.01-fold increase of cellular uptake at 60 hrs after treatment of 30 μM 6E1-conjugated liposome when compared with the control ILP (Figure 3). Alternatively, HIRMAb (a rabbit anti-insulin receptor monoclonal antibody)-conjugated liposome encapsulating plasmid encoding β-galactosidase was generated, and intravenously injected to Rhesus monkey to demonstrate global expression of β-galactosidase in the primate brain [110,111]. Given the nearly impermeable nature of the BBB, successful drug delivery to the brain through receptor-mediated transcytosis and subsequently to the neurons through endocytosis promises to unlock innovative advancements toward ameliorating neurodegenerative diseases.

## 9. Future Directions

A common theme among optimized ITs, ADCs, and ILPs is the quality of monoclonal antibody, and there will always be a need to generate the antibody that is translatable to the clinic. As described in this review, a proper screening strategy to pinpoint the antibody of interest is critical. Our cell-based IT screening system closely resembles the structural compositions of ITs and ADCs. As we expand our knowledge on critical features associated with methodological efficiencies and therapeutic efficacies, we can design and modify the screening strategies accordingly including inclusions of antibody phage display and single B cell antibody technologies. These advancements allow us to evaluate multiple feasibilities to maximize the potentials of the antibodies including, but not limited to, ITs, ADCs, and ILPs.

## 10. Conclusions

In this review, we described a current status of ITs, including our application of ITs as a cell-based IT screening system to obtain monoclonal antibodies that are suitable for further development. Our screening system, which exploits unique characteristics of antibody and engineered toxin DT3C, is a strategy to identify and isolate hybridomas that produce monoclonal antibodies that bind to cell surface molecules and are internalized into the cells to induce cytotoxicity. This system, both rapid and direct, is a platform for discovery of novel IT-, ADC-, and ILP-compatible monoclonal antibodies in vitro.

## Figures and Tables

**Figure 1 toxins-12-00658-f001:**
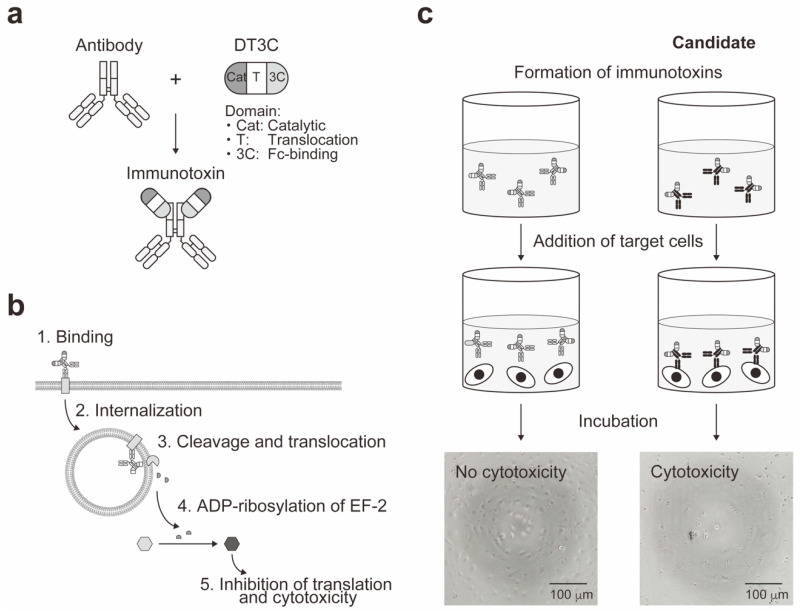
Mechanism of antibody: DT3C IT. (**a**) Formation of DT3C-mediated IT. DT3C consists of catalytic (Cat), translocation (T), and Fc-binding (3C) domains. The Fc-binding domain of DT3C specifically binds to an antibody. (**b**) Mechanism of IT-induced cytotoxicity. IT initially binds to an antigen expressed on the cell surface, and internalized into the cell where translocated terminus of DT3C is cleaved by the cellular furin protease, and catalytic domain of DT3C is released into the cytoplasm. Consequently, the catalytic domain ADP-ribosylates EF-2 and inhibits the protein translation machinery. (**c**) Cell-based IT screening system. Inside a well of the cell culture plate, antibodies secreted into the supernatant of the hybridoma library are pretreated with DT3C to form ITs. Subsequently, target cells are transferred into the well, and incubated. If the IT is bound to the target cells and internalized, then this leads to inhibition of protein translation machinery and ultimately cytotoxicity.

**Figure 2 toxins-12-00658-f002:**
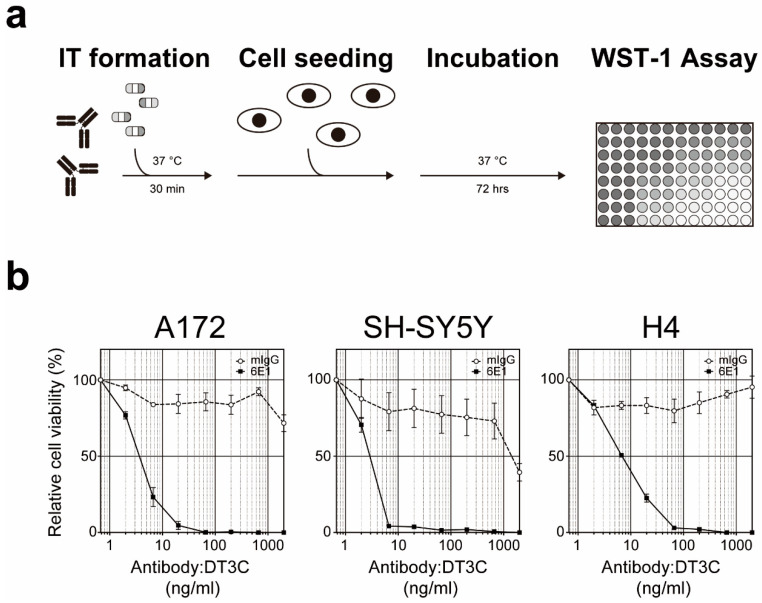
Cytotoxic activity of 6E1:DT3C IT. (**a**) Schematic diagram of IT assay. Purified 6E1 or control mouse IgG (mIgG) antibodies were pre-incubated with DT3C at 37 °C for 30 min to form ITs. After the IT formation, the indicated cells were seeded with various concentrations of ITs (*n* = 3 per treatment) and incubated at 37 °C for 3 days. Cell viability was measured by using WST-1 reagent. (**b**) Relative cell viability of A172, SH-SY5Y, and H4 cell lines after 6E1:DT3C treatment. The 6E1:DT3C IT induced cytotoxicity in all three cells lines tested whereby the logEC50s (ng/mL) were 4.59 (A172), 2.27 (SH-SY5Y), and 6.87 (H4). Assuming that 2 DT3Cs (75 kDa each) bind to one antibody, antibody:DT3C at 2000 ng/mL corresponds to 13 nM. Representative results of triplicate independent experiments. Data represent AVG ± SD.

**Figure 3 toxins-12-00658-f003:**
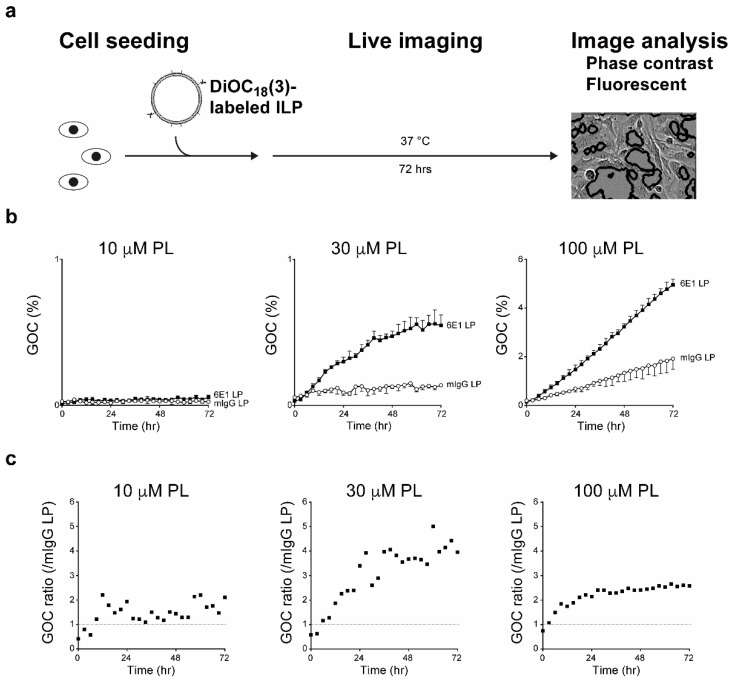
Cellular uptake of 6E1-conjugated liposome. (**a**) Schematic diagram of cellular uptake procedure. Liposomes were initially generated through dissolution of lipids in ethanol, injection of the ethanol solution into aqueous buffer, and extrusion through polycarbonate membranes. Following the extrusion, the liposomes were conjugated with 6E1 or control mouse IgG (mIgG) antibodies, and fluorescently labeled with DiOC_18_(3). A172 cells and ILPs were simultaneously transferred to a 96-well plate. The cells were treated with the indicated phospholipid (PL) concentrations of ILPs (*n* = 3 per treatment), and incubated at 37 °C for 3 days. Phase contrast and fluorescent images were acquired by using IncuCyte ZOOM (Essen BioScience, Inc.; Ann Arbor, MI, USA). Green fluorescent areas were determined as percentage of green object confluency (GOC). (**b**) Enhanced uptake of 6E1-conjugated liposome. At 100 μM, 6E1-conjugated liposome was significantly taken up by the cells, and mIgG-conjugated liposome also demonstrated gradual increase of cellular uptake. At 30 μM, 6E1-conjugated liposome demonstrated recurrent increased cellular uptake while the control remained relatively low. At 10 μM, cellular uptake of both ILPs remained constantly similar. Representative results of triplicate independent experiments. Data represent AVG ± SD. (**c**) Ratio of GOC between 6E1-conjugated and control mIgG-conjugated liposomes. If there was no difference in cellular uptake between these ILPs, then the ratio would remain at 1 as indicated by dotted lines. The most enhanced difference was observed at 30 μM liposomal concentration wherein 6E1-conjugated liposome exhibited 5.01-fold (60 hrs) increase of the cellular uptake when compared to the control. Representative results of triplicate independent experiments.

**Table 1 toxins-12-00658-t001:** FDA-approved cytotoxins, immunotoxin, and antibody drug conjugates.

Drug Name	Targeting Moiety	Toxin Moiety	Tumor Type	Approval Year	References
Cytotoxins					
Denileukin diftitox (Ontak^®^)	IL2	DT (DAB389)	CTCL	1999	[16]
Tagraxofusp-erzs (Elzonris^®^)	IL3	DT (DAB389)	BPDCN	2018	[18]
Immunotoxin					
Moxetumomab pasudotox (Lumoxiti^®^)	Anti-CD22 dsFv	PE (PE38)	HCL	2018	[19]
Antibody Drug Conjugates					
Gemtuzumab ozogamicin (Mylotarg^®^)	Humanized anti-CD33 mAb	Ozogamicin	AML	2000-approved 2010-withdrawn 2017-reapproved	[21]
Brentuximab vedotin (Acetris^®^)	Chimeric anti-CD30 mAb	MMAE	ALCL, HL, PTCL	2011	[22,23]
Trastuzumab emtansine (Kadcyla^®^)	Humanized anti-HER2 mAb	DM1	HER2^+^ BC	2013	[24,25]
Inotuzumab ozogamicin (Besponsa^®^)	Humanized anti-CD22 mAb	Ozogamicin	ALL	2017	[26]
Polatuzumab vedotin (Polivy^TM^)	Humanized anti-CD79B mAb	MMAE	DLBCL	2019	[27]
Enfortumab vedotin (Padcev^TM^)	Human anti-nectin-4 mAb	MMAE	UC	2019	[27]
Trastuzumab deruxtecan (Enhertu^®^)	Humanized anti-HER2 mAb	Deruxtecan	HER2^+^ BC	2019	[27]
Sacituzumab govitecan (Trodelvy^TM^)	Humanized anti-Trop-2 mAb	SN-38	Triple-negative BC	2020	[27]

ALCL: anaplastic large cell lymphoma; ALL: acute lymphoblastic leukemia; AML: acute myeloid leukemia; BC: breast cancer; BPDCN: blastic plasmacytoid dendritic cell neoplasm; CTCL: cutaneous T-cell lymphoma; DLBCL: diffuse large B cell lymphoma; DM1: derivative of maytansine 1; DT: diphtheria toxin; HCL: hairy cell leukemia; HER2: human epidermal growth factor receptor 2; HL: Hodgkin lymphoma; mAb: monoclonal antibody; MMAE: monomethyl auristatin E; PE: *Pseudomonas* exotoxin A; PTCL: peripheral T-cell lymphoma; SN-38: active metabolite of irinotecan; UC: urothelial cancer.

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
