# Peer review of "Immunotoxin Screening System: A Rapid and Direct Approach to Obtain Functional Antibodies with Internalization Capacities"

_toxins, 2020, doi:10.3390/toxins12100658_

Round 1

Reviewer 1 Report

This current article summarizes a current status of immunotoxins including the application of a cell-based screening system to isolate monoclonal antibodies that are suitable for further development as immunotoxins (IT), antibody drug conjugates (ADC) and immunoliposomes. Toxins havebeen engineered to develop potent therapeutics cytotoxins andimmunotoxins, which are  modalities with highly selective targeting capabilities.

This is the future therapy for a platform of selective pathologiesand this screening strategy permits identification of monoclonalantibodies that recognize cell-surfcae moleculs, selections of theantibodies that are internalized into cells, the antibodies thatinduce cytotoxicity and the posibillity to determin the state-specific activitiy of the antibodies. It is very important to have a therapy as more specific as we can and this cell-based IT screening system which utilizes distinct features of antiboy andengineered toxin, is a rapid, and perhaps more impotantly, direct method to indetify antibodies that recognize cell surfacemolecules and are internalized into the cells to induce cytoxicity.

As you pointed in your article there are three major steps to generate hybridoma library: immunization, cell-fusion of splenocytes and myeloma cells and library constriction of hybridomas through hypoxanthine-aminopterin-thymidine (HAT) selection. ADC (antibody drug conjugates) consists of a monoclonal antibody that is linked to cytotoxic payloads by non-cleavable or cleavable linker. In this situation the quality of the monoclonal antibody is crucial because therapeutic property of ADC is partially related to the characteristics of its antigen.The information that you startified in your article about ADC it must be promoted beacuse it has an important role in theefficacies and side effects of the cell-antibody theraphy. A critical feature is DAR (drug-to-antibody ratio) beacuse highDAR affects antibody structure and stability, and loqw DAR cand decrease efficacy; this aspect is very important because DAR values must be restricted to the manageable ranges.Monoclonal antibodies or their fragments are conjugated to the liposomal surface to generate ILPs (immunoliposomes) and a lack of increased anti-tumor activity after ILP administration has also been reported. ILPs it is also used in the delivery of the drugs across the blood brain barrier into the brain through receptor-mediated transcytosis and also to the neurons through endocytosis.

The screening system which exploits unique characteristics of antibody and engineered toxin is a very good strategy to identify and isolate hybridomas that produce monoclonal antibodies that bind to cell surface molecules and are internalized into the cells to induce cytotoxicity. This system is a platform for the future therapies based on monoclonal antibodies.

This present article is written in a clear and concise manner, purposing a new screening system that is capable to identify and isolate hybridomas that produce monoclonal antibodies, with precision and clarity, proving to be a launch platform in it’s field.

Author Response

Thank you very much for your thoughtful comments and careful assessment of our manuscript.

Reviewer 2 Report

The article entitled “Immunotoxin Screening System: A Rapid and Direct Approach to Obtain Functional Antibodies with Internalization Capacities” is a detailed analysis of the current status of immunotoxins including the application of a cell-based screening system to isolate monoclonal antibodies, suitable for further development as immunotoxins, antibody drug conjugates, and immunoliposomes.

This review is well written and represents an important addition to the literature.

On this basis, the paper deserves publication on Toxins journal after minor changes, as reported below:

-Check the references style in the text according to Instructions for Authors: reference numbers should be placed in square brackets [ ].

-Table 1 must be revised: Change “year approved” by “approval year”. Change “Refs” by “References”. Choose a heading also for the first column, such as “Name”. For “Indication” do you mean the “Tumor type”? If yes change by “tumor type” or “Type of cancer”. In order to be more clear, place the numbers “70-71” as well as “72-73” in the same line. Moreover, place reference numbers in square brackets [ ].

Author Response

Thank you very much for your thoughtful comments and recommendations.  Following your suggestion, we double-checked the reference style in the text, and now, reference numbers are placed in the square brackets.  Furthermore, we modified Table I, notably the column subheadings to include “Drug Name,” “Tumor Type,” “Approval Year,” and “References;” as well as reference numbers to be placed in the same line and in the square brackets.

Reviewer 3 Report

The review is well written and covers an important field of immunotherapy in which the success is greatly  dependent on functional antibodies with internalization properties.

Few minor points:

  1. Figures need to be higher resolution
  2. Figure 2a and 3a, rather than showing three arrows for 24hr incubation, can be changed to one 72 hrs incubation.
  3. Figure 2b, antibody-DT3C concentration should be in ng/ml.

Author Response

Thank you very much for your thoughtful comments and suggestions.  We changed Figs 2a and 3a to include one single arrow indicating 72 hr incubation, as well as Fig. 2b to illustrate antibody-DT3C concentration in ng/ml.  Accordingly, we modified the text (page 6, lines 46-47) and figure legend (page 7, lines 6-8) to reflect the change in concentration.  Additionally, we included the figures with higher resolution.